# First evidence for an aposematic function of a very common color pattern in small insects

Rebeca Mora-Castro[1,2,3]*, Marcela Alfaro-Córdoba[4,5], Marcela Hernández-Jiménez[2,6], Mauricio Fernández Otárola[3,7], Michael Méndez-Rivera[8], Didier Ramírez-Morales[8], Carlos E. Rodríguez-Rodríguez[8], Andrés Durán-Rodríguez[9], Paul E. Hanson[3]

1 Centro de Investigación en Biología Celular y Molecular, University of Costa Rica, San José, Costa Rica, 2 Centro de Investigación en Ciencia e Ingeniería de Materiales, University of Costa Rica, San José, Costa Rica, 3 Escuela de Biología, University of Costa Rica, San José, Costa Rica, 4 Centro de Investigación en Matemática Pura y Aplicada, University of Costa Rica, San José, Costa Rica, 5 Escuela de Estadística, University of Costa Rica, San José, Costa Rica, 6 Escuela de Física, University of Costa Rica, San José, Costa Rica, 7 Centro de Investigación en Biodiversidad y Ecología Tropical (CIBET), University of Costa Rica, San José, Costa Rica, 8 Centro de Investigación en Contaminación Ambiental (CICA), University of Costa Rica, San José, Costa Rica, 9 Protolab, University of Costa Rica, San José, Costa Rica

* rebeca.mora@ucr.ac.cr

**Data Availability Statement:** All data and code files are available from the public repository: https://malfaro2.github.io/Mora_et_al2/.

## Abstract

Many small parasitoid wasps have a black head, an orange mesosoma and a black metasoma (BOB color pattern), which is usually present in both sexes. A likely function of this widespread pattern is aposematic (warning) coloration, but this has never been investigated. To test this hypothesis, we presented spider predators (*Lyssomanes jemineus*), both field-captured and bred in captivity from eggs, to four wasp genera (*Baryconus*, *Chromoteleia*, *Macroteleia* and *Scelio*), each genus being represented by a BOB morphospecies and black morphospecies. We also used false prey, consisting of lures made of painted rice grains. Behavioral responses were analyzed with respect to presence or absence of the BOB pattern. In order to better understand the results obtained, two additional studies were performed. First, the reflection spectrum of the cuticle of the wasp and a theoretical visual sensibility of the spider were used to calculate a parameter we called "absorption contrast" that allows comparing the perception contrast between black and orange in each wasp genus as viewed by the spider. Second, acute toxicity trials with the water flea, *Daphnia magna*, were performed to determine toxicity differences between BOB and non-BOB wasps. At least some of the results suggest that the BOB color pattern may possibly play an aposematic role.

## Introduction

Many small (<10 mm) parasitoid wasps have a black head, an orange (or reddish orange) mesosoma and a black metasoma (BOB color pattern). This color pattern has been found in species belonging to 23 families of Hymenoptera (including phytophagous sawflies), and is especially common in neotropical scelionid wasps (Platygastridae; formerly Scelionidae), but is also common in evaniid wasps from diverse biogeographic regions; moreover, this color

**Funding:** This work was funded by the University of Costa Rica, project 801A5B50. The funders had no role in study design, data collection and analysis, decision to publish, or preparation of the manuscript.

**Competing interests:** The authors have declared that no competing interests exist.

pattern is usually present in both sexes [1]. In previous research it was found that the spectral blue components of the orange and black color in scelionid wasps are almost identical, suggesting that there is a common compound for the pigments [2], but the identity of the pigment remains unknown.

Conspicuous coloration such as the BOB pattern is often, but not always, indicative of aposematism [3], whereby predators learn to associate particular color patterns with noxious chemical defenses, although this learning process is much more complex than simply developing an aversion to specific types of prey [4]. In some larger insects contrasting black and orange color patterns are known to serve as aposematic (warning) coloration for potential predators, mostly vertebrates [5], and it is possible that the BOB pattern serves as a warning pattern for smaller (invertebrate) predators, although this has not yet been tested.

To test whether the BOB pattern also serves as aposematic coloration for predatory invertebrates, we chose a common (but understudied) jumping spider (Salticidae) for our experimental trials. Molecular and electrophysiological data suggest that color vision in the principal eyes of most jumping spiders is based on only two types of photosensitive pigments, one sensitive to ultraviolet (UV) light, the other to green light, although a few of them also exhibit filter-based trichromacy [6, 7]. Jumping spiders exhibit the following pairs of eyes: anterior median eyes (spatial acuity and depth of field), posterior lateral eyes (lower spatial acuity and wide field of view), anterior lateral eyes (high-resolution spatial vision due to densely packed receptors with forward-facing acute zones) and side-facing posterior median eyes (reduced in most genera). These characteristics of salticid spiders have allowed them to achieve the highest visual acuity thus far measured in any arthropod, with the ability to classify prey into categories [8–10].

Among the various jumping spiders present in our collecting sites, we chose to use *Lyssomanes jemineus* Peckham & Wheeler, because it was one of the most common species. *Lyssomanes*, like most salticids, are hunting spiders, primarily insectivorous, and behavioral observations suggest a strong role for vision in predation, although visual acuity in *Lyssomanes* may be somewhat lower than that of other salticids, such as *Portia* [11]. *Lyssomanes* species are diurnal foliage dwellers and two hunting behaviors predominate: a sit-wait strategy followed by springing from the underside of the leaf to the upper surface (they usually sit on leaves that are exposed to the sun waiting to ambush prey on the upper surface), and a hunting behavior that consists of exploring both sides of the leaf, actively searching for passing insects [12]. Thus, *Lyssomanes*, like most salticids, can be characterized as a hunting spider that does not use webs, and which executes behavioral responses such as: prey detection by sight, stalking, and attacking by jumping towards the prey [13, 14]. Penultimate juveniles and adults of *Lyssomanes* usually hunt and feed on soft-bodied insects such as dipterans, small hemipterans, and small wasps [12].

The objective of this study was to determine if the BOB color pattern in small parasitoid wasps is aposematic, through toxicity tests and behavioral analysis of invertebrate predators that depend on vision to capture prey. Our hypothesis is that the BOB pattern in scelionids and other small wasps is aposematic, acting as a signal for vision-dependent invertebrate predators. We had four predictions: 1) all-black prey would be preferred over prey with the BOB pattern; 2) experienced predators (collected in the field) would avoid the BOB pattern more than inexperienced adults reared in captivity; 3) greater perceived contrast between black and orange colors would trigger fewer attacks from the predator, since this would amplify the aposematic BOB pattern; 4) when comparing wasp species with the BOB color pattern with all black species, the former would show greater toxicity.

## Materials and methods

### Scelionid collection

The collection site with the greatest number of scelionid wasps of interest was in the Central Valley of Costa Rica, in a forest patch within the Protected Zone of El Rodeo, which is located near Ciudad Colón, San José (9˚52'–9˚56'N, 84˚14'–84˚20'W, 650 m.a.s.l.). The area is composed of secondary forest and a remnant of primary forest (approximately 200 ha). A collecting permit was obtained from the National System of Conservation Areas (Ministry of Environment and Energy), document N˚ ACTo-050-18, and all experimental protocols were carried out in accordance with our institution's guidelines and regulations. Because the hosts of scelionid wasps (insect eggs) are very difficult to locate, live wasps were obtained via sweeping with insect nets which requires considerable time and effort. Between March 2017 and November 2018 thirty trips were made to the site, and on each visit three people working for about six hours obtained four to six scelionid wasps. The latter were promptly used in trials (see below) carried out in a makeshift laboratory at the same site. Within the wasp genera showing the black-orange-black (BOB) pattern, only some species have this pattern, the other species usually being totally black (Fig 1). The wasps were identified by R.M. and P.H. using keys to scelionid genera [15]; voucher specimens were deposited in the Museum of Zoology at the University of Costa Rica. Four genera were collected from this site and used in the experiments: *Baryconus* Foerster, *Chromoteleia* Ashmead, *Macroteleia* Westwood and *Scelio* Latreille. All four genera were represented by a BOB morphospecies and an all black morphospecies, providing a total of eight morphospecies. Due to the lack of taxonomic studies for three of the four genera, species level identifications were not possible. While there is a chance that some of the morphospecies included a mixture of more than one species, this problem was minimized (though not totally eliminated) by always collecting from the same site and carefully examining the specimens under a stereomicroscope.

### Salticid collection

*Lyssomanes jemineus* egg sacs (from which adults were reared in captivity; see S1 Appendix in S1 File), female adults, and juveniles were collected weekly from January 2017 to January 2019. Collecting was done in Finca 3 of the University of Costa Rica, San Pedro de Montes de Oca, San José, Costa Rica (9˚56'07"N, 84˚03'04"W, 1300 m.a.s.l.). This site is located in a disturbed urban environment which exhibits mainly semi-woodland vegetation of native and introduced herbaceous plants [16]. This locality has an average annual precipitation of 1200–1500 mm and an average temperature of approximately 21˚C. Spiders and egg sacs were collected from 0 to 1.5 m above ground level. Since many of the plants were quite widely spaced, the transect consisted of 25 to 30 locations where diverse herbaceous plants were haphazardly searched. For the collection of egg sacs weekly samplings of approximately three hours each were carried out from May to September, according to the reported phenology for this species [12]. Each egg sac (usually located on the underside of the leaf) was collected manually without being detached from the leaf and was quickly transferred to the laboratory using 15 ml sterile Falcon tubes with a humid cotton ball. For the collection of adults, the same methodology was applied, but mainly from April to October. Specimens were identified by G.B. Edwards.

### Observations of spiders with live prey

Field-collected juvenile and adult spiders, as well as adults reared from egg sacs in the laboratory (for about a year), were observed one at a time in an experimental arena (see below). Adult spiders reared in the laboratory had no prior contact with the wasps. Six types of

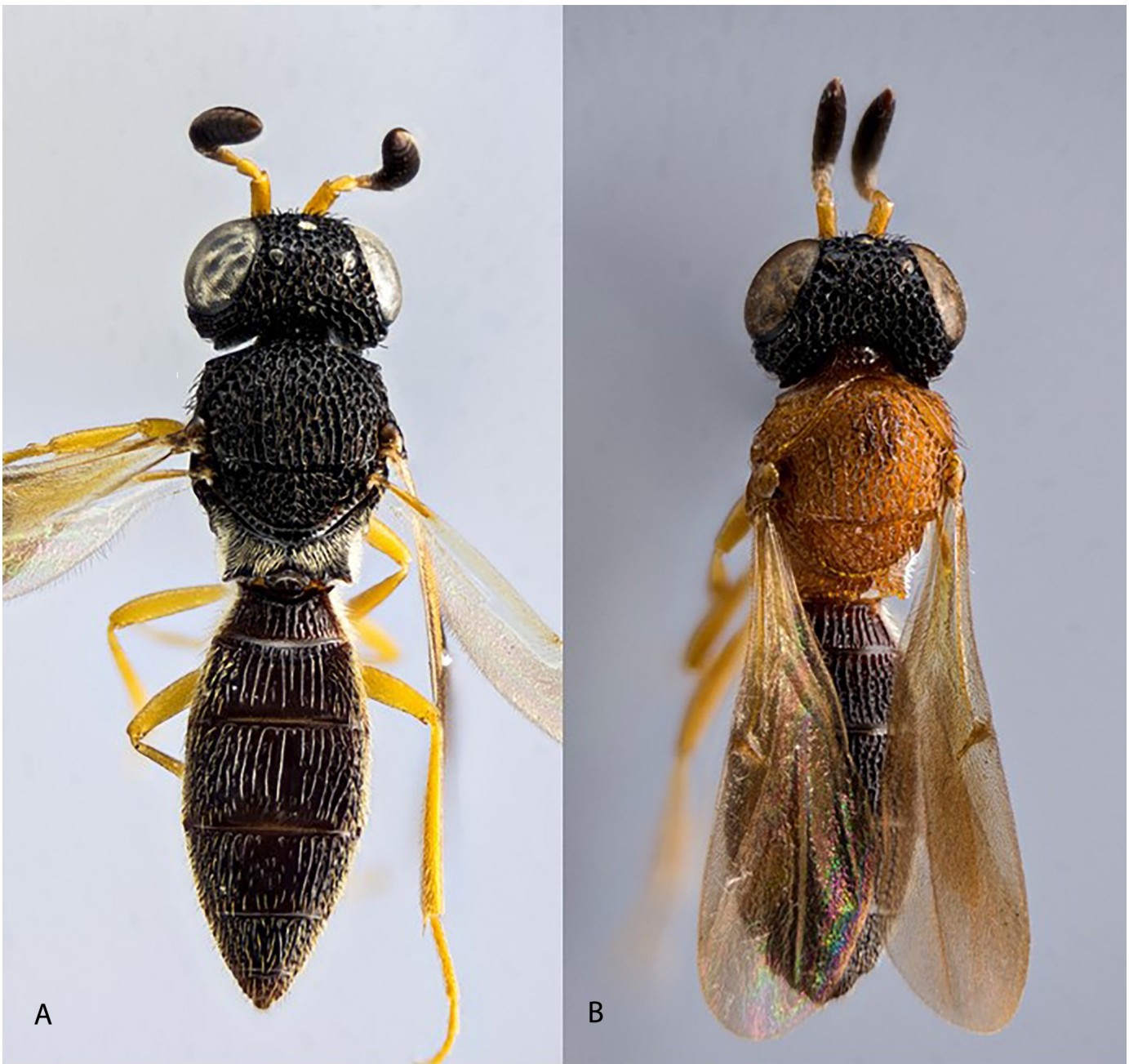

**Fig 1. Two scelionid wasp color patterns within the same genus.** Color patterns in the genus *Scelio*: black (A), and BOB (B). The focus-stacked macro photograph was obtained with a Reflex 850 camera coupled to a 20x microscope lens.

behavior were recorded: 1) prey detection as indicated by swivelling the cephalothorax; 2) following or stalking prey; 3) crouching, jumping and contacting the prey; 4) piercing and ingesting the prey; 5) prey detection followed by withdrawal from the prey; 6) prey undetected or ignored. In the results the first two were combined as "detect" (although detection was not always followed by stalking), three and four were combined as "attack" (although jumping was rarely followed by ingestion), and five and six were combined as "avoid".

To facilitate observations of such small specimens we used a closed transparent acrylic box with an internal volume of 1550 cm$^3$. The acrylic plates were made in a CNC cutting machine, model Redsail CM1690, with a red laser tube with 100 watts of power. To provide greater structural complexity and additional routes for approaching the prey, an acrylic, tree-like structure was placed in the center of the floor [17]. We also incorporated a white background on the floor and natural light was allowed to enter the remaining walls of the acrylic box. In order to emulate the natural environment as much as possible, all observations were carried out in a makeshift kiosk embedded in the forest in which the wasps were collected. Behavior was filmed with a Sony RX100 IV camera and was done in FullHD 1080 x 1920 at 60fps.

In order to minimize potential confounding factors, we used the following standardized conditions for all experiments with spiders.

1. We standardized spider sex (females only), age (more than 160 days old) and body size (considered in the statistical analysis).

2. An 8-day pre-test fast ensured that the test spiders would be motivated to feed during testing.

3. We used each prey and predator for just a single trial.

4. Each spider was allowed 60 s of acclimatization in the chamber, before introducing the prey.

5. Between tests, the chamber was washed with 80% ethanol, followed by distilled water, and then dried in order to eliminate any chemical trail of predator or prey [18–20].

6. Each trial finished when the prey was eaten or after 40 minutes of observation.

## Observations of spiders with false prey (rice lures) in the automated cage

Most of this research was based on live predators (salticids) and prey (scelionids). However, false prey consisting of painted rice lures were also used, since some previous studies have examined aposematic signals in insects by utilizing painted models as prey in order to achieve greater standardization with respect to potentially confounding factors such as posture, size and shape [21, 22].

The experimental arena, consisting of an automated cage for observing spiders with false movable prey (rice lures, explained below), was made of a closed transparent acrylic box with the same characteristics as the ones used for live prey. In order to study whether background colour affected the detectability of BOB or black rice lures, two types of background were used: one acrylic box had white-colored walls and the other had black-colored walls. All observations were carried out between 07:00 h and 11:00 h in a laboratory with ambient lighting from fluorescent ceiling lamps (photoperiod 12h:12h L:D).

In order to generate movement in the lures, the acrylic box was placed inside a frame that contained an automated mechanism (Fig 2). The rice lures were moved by a motor mechanism that allowed for two-dimensional movements. Each lure was connected to a moving part through a nylon thread, generating movement along two axes. This mobile part consists of a magnet on the outside of the acrylic box which in turn moves a magnet on the inside of the box. The control unit has programming functions that allow one to execute different patterns of movement and comprises an interaction interface with the user; for this study, we used only horizontal movements along the side walls of the box (attempting to simulate the movements of live scelionid wasps that we previously observed to be attractive to the spiders). For the programming of this movement pattern the Arduino platform was used.

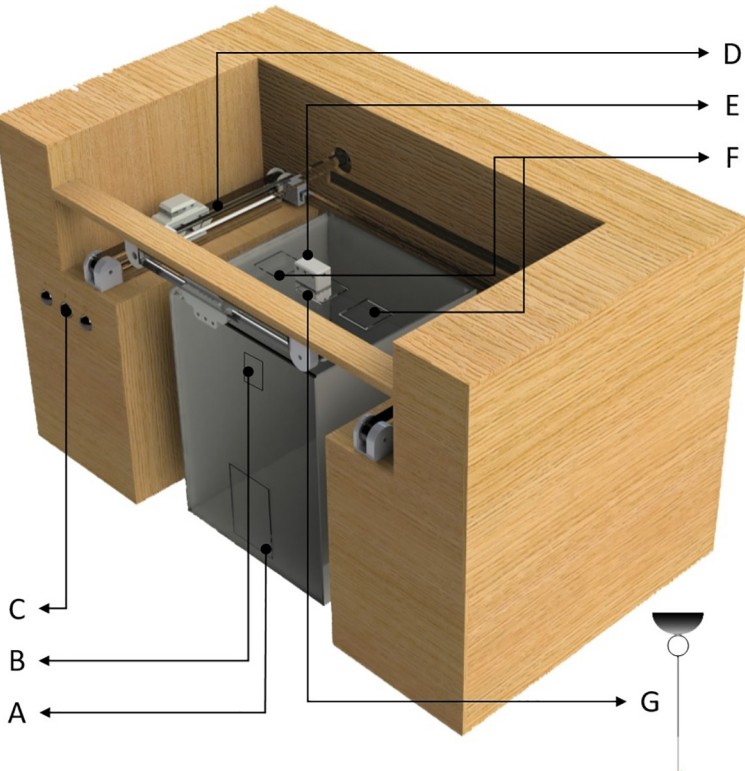

**Fig 2. Components of the automated cage.** Aperture for introducing the spider (A). Aperture for introducing live prey (B). Buttons controlling the moving lure (C). Motor mechanism consisting of 4 guide blocks, 4 rails and 2 means of movement that use two pairs of independent rails; each electric motor is connected by a toothed belt to a guide block, which in turn is connected to a busbar that moves the remaining guide block located in parallel; each pair of guide blocks located in parallel is connected to a moving part through a nylon thread, generating movement along an axis (D). Upper magnet (E). Aperture for introducing false prey (lure) (F). Lower magnet from which the lure is suspended by a transparent thread inside the acrylic box (G).

The interaction between live spiders, lures (made of painted rice) and the experimental arena consisted of the following:

a. Spiders and lures: during each trial an adult spider had simultaneous access to two types of lures, one painted in the BOB pattern and the other painted all black.

b. Lures and automated cage: The BOB and black lure moved continuously and simultaneously, one on each lateral wall of the arena, emulating live scelionid wasp movement. All lures were 4 mm in length, similar to that of the scelionid wasps. In order to objectively paint the grains, not based on the colors detected from our human visual space, but on the spectral characteristics of the BOB colors in wasps, the reflectance spectra of both wasps and painted rice grains were considered (S2 Appendix in S1 File). The spectral characteristics of BOB colors were compared with those of paints in order to choose the lures having spectral features most likely to interact with the visual system of the jumping spiders (S1 Fig in S1 File). With respect to the four scelionid genera included here, we chose to use the previously reported mean reflectance curves for *Baryconus* because this data set showed low dispersion compared with those for other scelionid genera [2].

c. Data collection: Each trial was carried out for 40 minutes. We recorded the following behaviors: a) spider detects prey, as indicated by swiveling its cephalothorax; b) prey stalked and/

or contacted; c) prey undetected or ignored; d) prey detected and spider withdraws; e) the color pattern (BOB or black) that is detected first. We also measured the body length of each spider, and noted whether a noticeable silk dragline was produced while jumping toward the prey (small amounts of silk produced while walking were not included).

## Absorption contrast parameter for the green spectral components of the BOB pattern

In order to elucidate whether the spider can distinguish between orange (BOB) and black, and to determine whether the orange color varies among genera, the method described in S2 Appendix in S1 File was also used to obtain spectral green components of the reflectance spectra of the black and orange color of the BOB pattern in the four wasp genera, as previously reported [2]. Since preliminary studies of other salticid spiders suggest the presence of photopigments with absorption bands in the UV (ca. 360nm) and the green (ca. 520nm) [23], we used a Govardovskii nomogram [24] for the normalized absorption spectrum of the $\alpha$ band of A1 type pigment with $\lambda$ max = 520nm. Nevertheless, this description is not complete since it lacks information about the UV visual sensibility.

After calculating the spectral green components, the areas under each curve were obtained numerically. This quantity represents how much of the light reflected by the cuticle of the wasp can be absorbed by the photosensitive pigment of the spider and therefore can be processed by its visual system. According to this criterion, the absolute value of the arithmetic difference between areas should be proportional to a higher or lower capability of detecting a difference between colors. The difference was calculated as the area for the green component of the black color minus the area of the green component of the orange color. Negative values therefore indicate a major absorption of electromagnetic input corresponding to orange, whereas positive values indicate a major contribution to the absorbed light coming from the black color. We refer to these differences as "absorption contrasts", in the sense that the only assumption we are making about the physiology of the visual system of the spider is the type of photosensitive pigment present and its maximum absorption wavelength (as described above). The calculations described here were performed using the reflectance curves reported in reference [2]. The standard deviation of the areas under curve from the green components were used for estimating an error for the calculation of the absorption contrast.

## Statistical analysis

There were three different statistical analyses in this work: (1) an experimental design with spiders and live prey, that uses a multinomial model to describe 136 trials, with each trial as the statistical unit, and the most common spider activity per trial as a response, (2) a statistical description of the timeline for the same trials, taking each spider action as the statistical unit, and (3) an experimental design with spiders and false prey in the automated cage, with 30 separate trials.

The first analysis consisted of a total of 136 trials, with three groups of spiders (68 field juveniles, 51 field adults and 17 captive adults) and live prey during which the following data were recorded: spider type (juvenile, field adult, captive adult), wasp color (BOB or black), spider behavior (behavioural scale explained before), time when the spider reacted (0–5, 5–10, 10–20, 20–30, 30–40 minutes), the distance between the spider and the prey during each behavior, presence/absence of a noticeable silk dragline, and wasp and spider body length. For each of the 136 trials consisting of a wasp with a spider, the size of each was measured (S2 Fig in S1 File). Although there is size variation within each group, predator (spiders) and prey (wasps)

fall within the same general range, with field-captured adult spiders being the largest (S2 Fig in S1 File).

A multinomial logistic regression [25] was fitted to a simplified response: the most common activity for each spider was classified as "detect", "attack" or "avoid", in order to correct for zero or near zero counts in each response category. The full model includes the following covariates: wasp color, spider type, wasp size, spider size, wasp genus and presence/absence of a noticeable silk dragline during jumping. This model was compared with simplified versions, with the result being that the model with wasp color and spider type was the model with the best fit, according to the AIC statistic.

Additionally, a second analysis was carried out with the same trial, where the number of individual actions for each spider was recorded during each of the four 10-minute time slots, and distinguished between the wasp colors they were acting on, in order to define a clear time-line of spider behavior in the controlled experiments. In this case we had 954 individual actions included in the 136 trials.

A third analysis of a separate experiment with lures in the automated cage included 30 trials, during which the following variables were recorded: presence of noticeable silk/absence of noticeable silk, spider size, background experimental arena color (black or white) and color of false prey (BOB vs black) that first attracted the spider.

The response variable in this case was constructed by registering whether *L. jemineus* responded in the same way to black and BOB lures, i.e. if it attacked, detected or avoided both the black and BOB lures, coded as 1, and whether *L. jemineus* responded in a different way to black and BOB, i.e. if it attacked black and detected the BOB, or if it detected the BOB and avoided the black lures, coded as 2. Contingency tables were calculated, along with a log likelihood ratio (G-test) independence test for the response versus each of the variables: presence/absence of a noticeable silk dragline, background color and prey color that first attracted the spider.

Statistical analyses were performed using R (version 3.6.2, R Core Team, 2019). Data formatting and figures were prepared using the Tidyverse packages [26]. Multinomial logistic regression was done using the nnet package [25].

## Wasp extract preparation for toxicity tests

To obtain the wasp extracts some modifications of Arenas research [5] were implemented. Pools of three organisms from the same genus and with the same color were placed in 1.5 ml centrifuge tubes; the initial weight of each pool was taken with a Mettler Toledo XP 205 analytical balance. Extraction was done by adding 0.5 ml of methanol (99.8% purity) to each pool of individuals and subsequent maceration was done with a glass pestle for 5 minutes. After maceration, the tubes were centrifuged in a Minispin plus Eppendorf centrifuge at 14500 rpm for 10 minutes. The supernatant was transferred into a 6 ml glass vial and the pellet was discarded. The methanol from the supernatant was evaporated to dryness with an Organomation brand nitrogen concentrator, model MULTIVAPTM (flow of 7 l/min at 28˚C). For testing, 1400 μl of reconstituted water was added to the concentrate and homogenized using a vortex; this extract was considered as 100%, then dilutions of 75%, 50% and 25% from the original extract were prepared with reconstituted water to a final volume of 0.5 ml. A negative control of the extract, lacking the wasp pool, was prepared following the same procedure.

## Acute toxicity test with *Daphnia magna*

The toxicity of seven wasp extracts corresponding to four scelionid genera and the two color patterns of interest was measured by an acute toxicity test with the water flea *Daphnia magna*

Straus, based on a similar method described by [5]. This organism is frequently used for toxicity tests due to its sensitivity to xenobiotics, wide distribution, short life cycle and ease of culturing in the laboratory. The water fleas were kept in reconstituted water ($MgSO_4$, $NaHCO_3$, KCl, $CaSO_4$ x 2 $H_2O$) [28], at 21 ± 2°C, with a photoperiod of 12:12, and fed with the green algae, *Raphidocelis subcapitata*. The culture medium was changed every week, and after five-weeks adult individuals were discarded and a new culture started from neonates. For the adapted acute toxicity test [27], ten 24-hour neonates were exposed to 0.5 ml of each different dilution (triplicates) of the wasp extract during a 48-hour period in a dark room at a constant temperature of 22°C. The test end point was mortality (immobilization), which was recorded at 24 h and 48 h. The data were used to calculate the mean lethal concentration (LC50), using the R software, version 3.5.3 (R Development Core Team 2014) and using the "drc" package [28]; LC50 was defined as the relative concentration of the extract (considering the pure extract as 100%) that causes a mortality of 50% in the water fleas.

### Acute toxicity test with *Vibrio fischeri*

The Microtox test detects inhibition of bioluminescence in the bacterium, *Vibrio fischeri*, using the Microtox® 500 Analyzer. For the analysis, the 2% Basic Test method recommended by the Microtox® software was employed. Briefly, the lyophilized bacteria (Microtox Acute Reagent) was reconstituted and exposed to four dilutions from the original 100% wasp extract, prepared with Modern Water Microtox Diluent®. Light emission by the bacteria in contact with the extract dilutions was measured at 0, 5, and 15 minutes; data analysis by the Microtox® software was employed to determine the EC50 value, defined as the relative concentration of the wasp extract (considering the pure extract as 100%) that causes 50% bioluminescence inhibition.

## Results

### Experiment of spiders with live prey

A summary of the most common action per each of the 136 trials included in the final multinomial model is presented in Fig 3. The number of spiders differs between groups, which is why the bars for captive adults are smaller than the ones for field adults and field juveniles.

The aforementioned model suggests that not producing a noticeable silk dragline is associated with an increased chance of detecting versus attacking (p = 0.0160), which makes sense

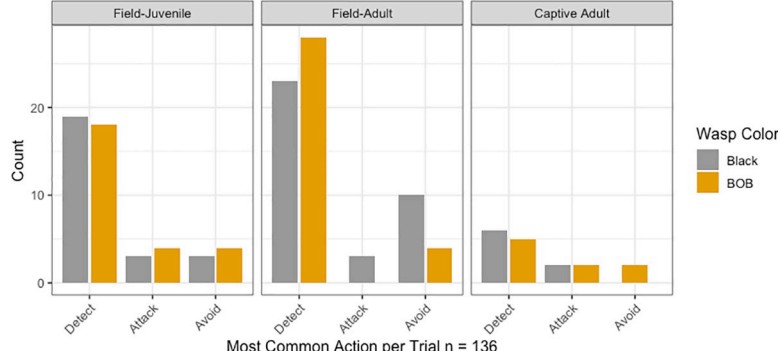

**Fig 3. Most common behaviours of *L. jemineus* in trials with different combinations of wasp color (BOB vs black) and spider type.** Each bar represents the proportion of spiders per type and wasp color (orange–BOB, gray–black), according to the most common action taken by the spider in each of the 136 trials.

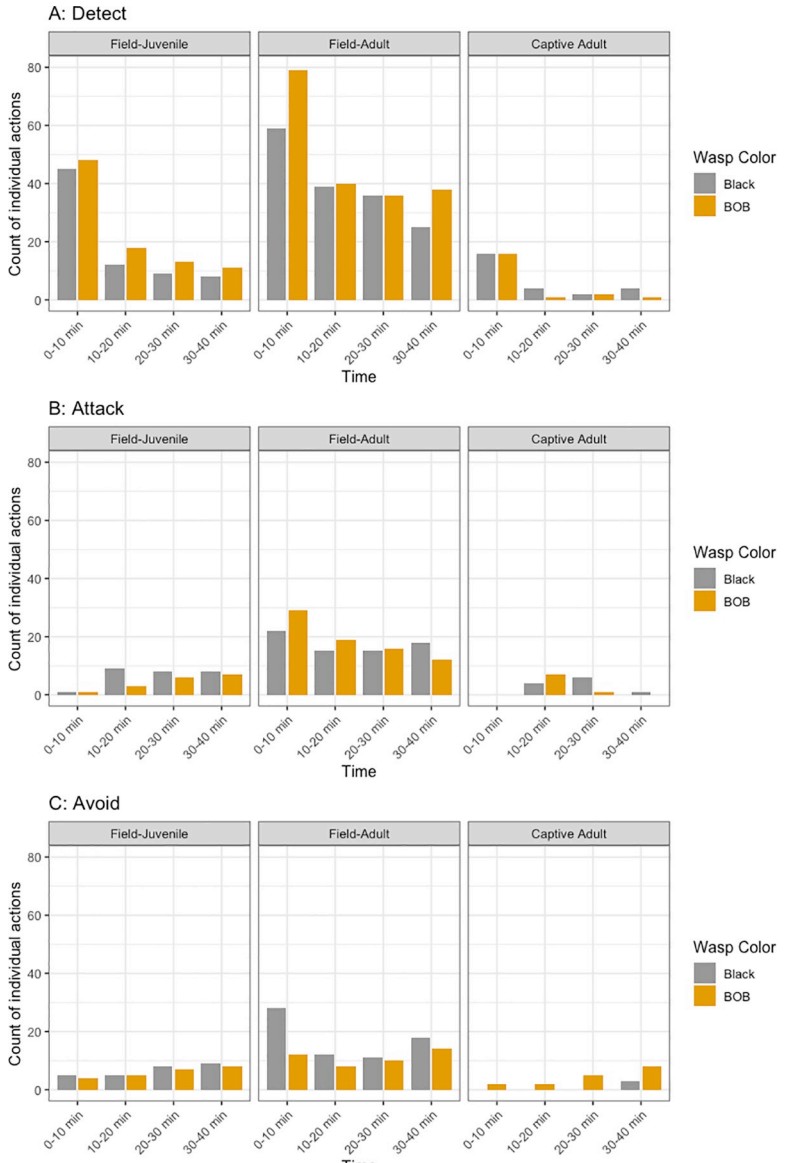

**Fig 4. Frequency of individual behavioral responses experienced by *L. jemineus* when confronted with live wasps.**
Behaviors are: detection (A), attacking (B), avoiding (C) All bar plots are represented according to behavior per time, spider type and prey color (black and BOB). Time slots without bars indicate that none of the spiders performed that action during that time slot. The total individual actions is n = 954.

since in many cases silk was more evident before attacking the prey (R. Mora-Castro, unpublished), an observation that warrants further research. Being a field-caught juvenile versus a captive adult did not make a difference in the most common spider actions. Other factors such as wasp color, wasp size, spider size, and wasp genus did not make a significant difference when included in the full model.

When the behavior of all spiders is analyzed individually and during the 40 min and in the different time slots a greater complexity of behaviors is observed, specifically in that the field adults usually differed from the captive adults. Fig 4 summarizes the 954 actions taken by all spiders throughout the timeline of the experiment, and data in Table 1 shows the detailed

**Table 1. *L. jemineus* individual behavioral responses.**

| Behavioral responses | Black | BOB | TOTAL |
|---|---|---|---|
| 1. Alert and swivel | 160 | 192 | 352 |
| 2. Follow or stalk | 99 | 111 | 210 |
| 3. Crouch, jump and contact prey | 99 | 101 | 200 |
| 4. Pierce and ingest | 8 | 0 | 8 |
| 5. Prey undetected or ignored | 83 | 48 | 131 |
| 6. Detect visually and withdraw | 14 | 37 | 51 |
| 7. None | 2 | 0 | 2 |
| TOTAL | 465 | 489 | 954 |

Frequency of individual behavioral responses experienced by *L. jemineus* when confronted with live wasps, with seven ungrouped behaviors, per wasp color. The total individual actions is n = 954.

count with the original individual actions. There are some time slots without observations such as 0–10 mins for captive adults in plot B. Thus, the number of spiders for each category changes according to each combination. It can be seen that captive adults detect/stalk wasps of both color patterns, but during certain time periods their behaviour is directed slightly more toward the black wasps than the BOB wasps (10–20 min and 30–40 min), a behavior not observed in field adults or field juveniles (Fig 4A). It is worth noting that captive adults attack only black wasps during the last time period, unlike the first 30 min during which both colors are attacked, as in field juveniles and adults (Fig 4B). Captive adults avoid wasps with the BOB pattern during most time periods, which differs from the behavior of field-caught spiders, which showed equal avoidance of both colors of wasps (Fig 4C).

There is a clear effect of experience, with captive adults showing a lower attack capacity than field spiders (including juveniles). This may be due to the fact that captive adults were reared on a simple diet and were thus responding to unfamiliar prey. However, there is some evidence for an innate response by captive adults, in avoiding attack responses and aversion to BOB wasps (Fig 4C).

## Green spectral components of the BOB pattern

The mean green spectral components of the black and orange colors of the BOB pattern in the four wasp genera are shown in Fig 5. In general, the spectral components for black and orange colors differ in the position of the maximum wavelength and the height of the peak, indicating different interaction of the reflected light with the photosensitive pigment. The exception is *Baryconus*, for which the spectral components of black and orange colors are similar. The absorption contrast parameters, calculated as described in the Methods section, together with the results of the predation experiments, are presented in Fig 6. The four genera were ranked from the highest negative value to the highest positive value of absorption contrast between black and orange spectral components. Also, the mean green spectral components of the black and orange colors of the BOB pattern from Fig 5, used for the calculation of the absorption contrast, were included as a visual aid. The standard deviation for the areas under the curve, used for estimating the error in the calculation, are shown as percentages inside square brackets. The observed events, classified in the categories "detect", "avoid" and "attack", as explained in the Methods section, are presented as percentages calculated with respect to total number of trials for each genus in the case of BOB specimens, and the total number of trials for black specimens of all genera. This choice is based on a conclusion of previous work of our group [2], which suggests that black color is spectrally similar between genera.

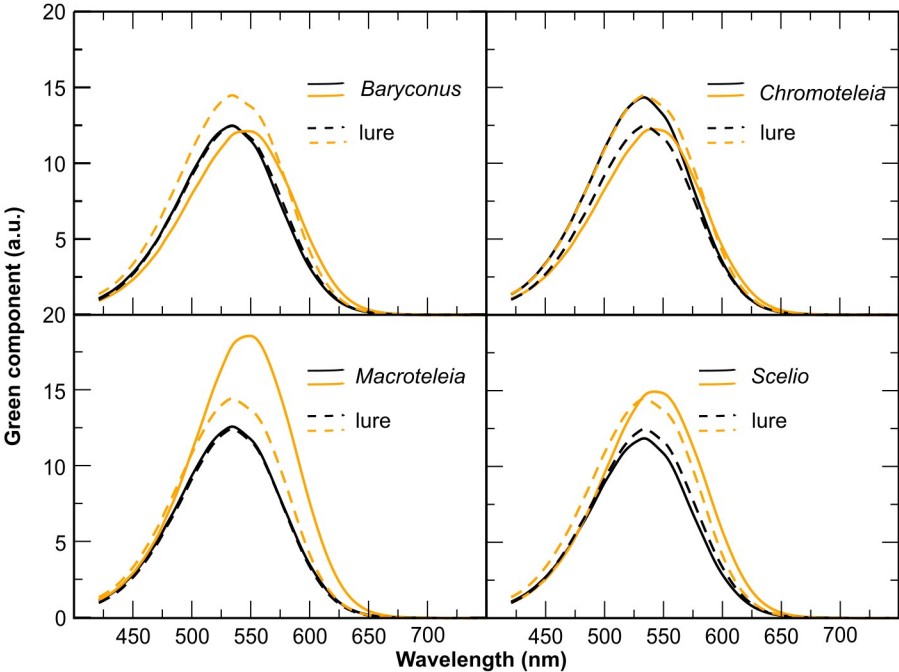

**Fig 5. Mean green component of the reflection spectra of both black and orange colors for both the cuticle of each genus and the blend chosen for painting the false prey (painted rice grain).** Green component, in arbitrary units (a. u.), of both black (black line) and orange (orange line) colors for the BOB color pattern of four genera of scelionid wasps (solid lines) and the paint chosen for the false prey (dashed lines).

The information in Fig 6 shows that for the wasps with a BOB color pattern, most of the "attack" events occurred with BOB-colored *Baryconus*, which also has the lowest absorption contrast. On the other hand, the greatest number of "detect" events occurred with BOB-colored *Macroteleia*, which has the highest absorption contrast and the highest contribution of orange, followed by *Chromoteleia* which has the second highest contrast value in absolute numbers, while lower values of black contribution, as in *Scelio* and *Baryconus*, yielded a lower percentage of "detect" events. The greatest number of "avoid" events occurred with BOB-colored *Scelio*, which has the second highest absolute absorption contrast.

## Experiment of spiders with false prey in the automated cage

Table 2 presents the results from the experiment with false prey. Each contingency table was tested for independence, and there was no evidence of dependence for background (G = 2.0879, p-value = 0.1485), color (G = 0.06728, p-value = 0.7953), or presence of noticeable silk (G = 0.0089311, p-value = 0.9247). In other words, spider behaviors are not associated with background color (black or white walls in the arena), the color of lure that was detected first, or the presence or absence of a noticeable silk dragline.

## Acute toxicity tests with *Daphnia magna* and *Vibrio fischeri*

During the exposure period, acute toxicity trials with *D. magna* resulted in a higher mortality (LC50 < 65.2%) of water fleas when exposed to the extracts from the BOB wasps, which employed *Chromoteleia* and *Baryconus* (Table 3). In the case of the black wasps, three out of

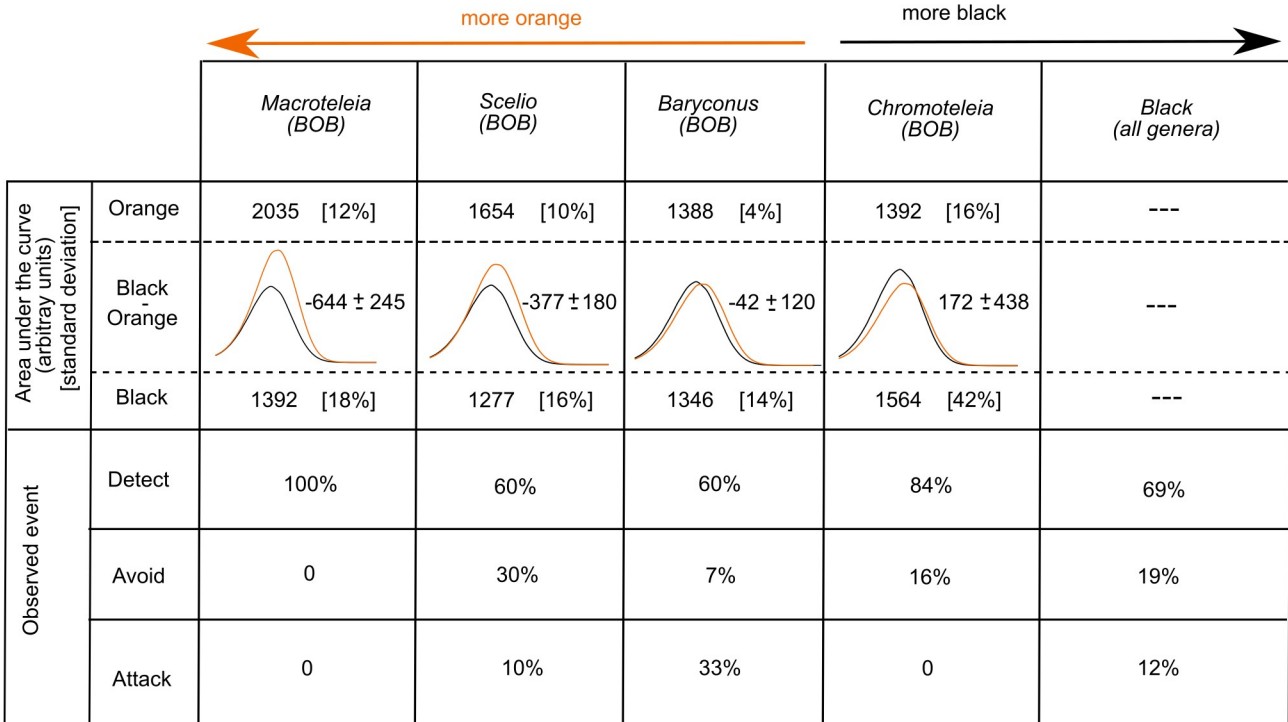

**Fig 6. Relationship between the green components of the BOB colors of each scelionid genus and the percentage of behavioral responses by *L. jemineus*.** Summary of information about the relationship between the green components of the BOB colors of each scelionid genus and the percentage of behavioral responses by *L. jemineus* observed and grouped in the categories "detect", "avoid" and "attack". The mean BOB color pattern areas under the curve, in arbitrary units, were calculated and compared for the green component of the black and orange colors, respectively (see also Fig 5). Values within square brackets represent the standard deviation of the areas.

four extract samples did not show toxic effects (LC50 > 100%) on *D. magna*; the one sample of black *Macroteleia* that showed toxicity had a lower mortality than with the BOB wasps. No mortality of water fleas was observed with the negative control, thus suggesting that the toxic effects are due to the components of the wasps. In the toxicity tests with *V. fischeri* only black *Macroteleia* and BOB *Chromoteleia* resulted in high bioluminescence inhibition. The results for the two genera have overlapping confidence intervals, so no difference can be inferred; on the contrary, there were no inhibitory effects with BOB *Baryconus* or black *Scelio* (Table 3).

**Table 2. Results from experiment with false prey in three contingency tables.**

| Behavioural responses | (A) | | (B) | | (C) | |
|---|---|---|---|---|---|---|
| | Background | | Color | | Presence of a silk dragline | |
| | White | Black | Black | BOB | No | Yes |
| *L. jemineus* took the same actions with both black and BOB wasps | 13 | 8 | 15 | 6 | 16 | 5 |
| | 43.3% | 26.6% | 50.0% | 20.0% | 53.3% | 16.6% |
| *L. jemineus* took different actions with black and BOB | 3 | 6 | 6 | 3 | 7 | 2 |
| | 10.0% | 20.0% | 20.0% | 10.0% | 23.3% | 6.6% |
| TOTAL (30) | 16 | 14 | 21 | 9 | 23 | 7 |

A total of 30 observations were aggregated according to the response variable (same or different actions) and three variables: (A) background color (white or black), (B) color (black or BOB) that first attracted the spider, (C) presence of a noticeable silk dragline while jumping toward the prey (yes or no). The counts and the percentage of total cases are indicated.

**Table 3. Toxicity (± C.I.) of extracts of scelionid wasps as determined in acute tests in the bioindicators *Daphnia magna* and *Vibrio fischeri*.**

| Wasp genus and color (pools of three specimens) | Total mass (mg) | LC50[a] (%[c]) (*Daphnia magna*) | EC50[b] (%[c]) (*Vibrio fischeri*) |
|---|---|---|---|
| *Baryconus* (BOB[d]) | 3,41 | 59.6% (50.2, 68.9) | > 100% |
| *Chromoteleia* (BOB) | 3,46 | 60.6% (51.5, 69.7) | 1.7% (0.9, 3.0) |
| *Chromoteleia* (BOB) | 3,75 | 65.2% (54.4, 75.9) | 5.2% (0.8, 32) |
| *Macroteleia* (black[e]) | 1,34 | 85.9% (73.5, 98.3) | 1.0% (0.4, 2.0) |
| *Macroteleia* (black) | 1,42 | > 100% | 2.1% (1.1, 4.1) |
| *Scelio* (black) | 0,66 | >100% | > 100% |
| *Scelio* (black) | 0,86 | >100% | > 100% |

[a] LC50: mean lethal concentration

[b] EC50: mean effect concentration

[c] %: Refers to the % of original extract dilution; 100% corresponds to the pure extract.

[d] BOB: black-orange-black color pattern

[e] black color pattern

Overall, the wasp extracts showed a much higher toxicity for the bacteria bioindicator (*V. fischeri*) than for the arthropods (*D. magna*).

## Discussion

### Evidence for an aposematic function of BOB coloration

A previous study showed that the black-orange-black (BOB) color pattern is very widespread among small parasitoid wasps and that it is usually present in both sexes [1], suggesting that this color pattern is not related to sexual behavior. The principal hypothesis regarding the function of this widespread color pattern is aposematism and to the best of our knowledge, the present research is the first to test this hypothesis. Although the results were mixed, at least two of our findings provide evidence that BOB coloration is indeed aposematic. First, although the behavioral responses were quite variable, we never observed a spider consuming a wasp with BOB coloration (Table 1), but we did observe a few instances of black wasps being ingested. Second, toxicity, a well-studied and common attribute utilized as a defense by aposematic prey [29], was determined in acute toxicity tests with *Daphnia magna* and showed that wasps with BOB coloration caused greater mortality than did the black wasps. This suggests that BOB wasps contain defensive compounds unlike black wasps, which would explain why only black wasps were consumed, but this requires further investigation. It is also possible that the BOB coloration acts synergistically with repellent odors, which has been shown to affect foraging behavior in another jumping spider [30].

It was also observed that the wasp extracts were more toxic for the bacteria (*V. fischeri*) than for the crustacean (*D. magna*). Unfortunately, although in the same toxicity range, no clear differences were observed for the extract samples that showed toxicity in the *V. fischeri* assays, due to overlapping confidence intervals. Nonetheless, results from this test should be taken into account when considering the potential presence of toxic compounds in the wasp extracts. A consistent result for both bioindicators is that the black *Scelio* did not show toxicity, and might be considered the least toxic of the four genera. The results for *D. magna* are more relevant for this study since differences between the wasp genera can be observed and because it is an arthropod as is the spider. Moreover, the extracts from the BOB wasps all showed toxicity

while just one out of four black wasp extracts did so; moreover, the EC50 values for the BOB wasp extracts were lower, i.e. had higher toxicity.

## Predator responses towards BOB wasps

It should be emphasized that considerable effort is needed to obtain live scelionid wasps; a total of about 540 hours were required to collect the wasps used in our experiments and even with this effort there were insufficient numbers for some of the trials. This limitation was one of the motivations for carrying out trials with lures. Despite considerable care taken in providing lures that closely matched the black and orange colors of the wasps, not subjectively through our visual perception but through spectrophotometric measurements, and an arena in which the movements of the lures were carefully controlled, the results were inconclusive. The likely explanation is that the lures lacked additional visual cues (e.g. legs and eyes) and/or chemical or behavioral cues necessary to elicit a predatory response, which has been shown in other jumping spiders [6]. Also, the absorption contrast of the paint chosen is somewhere between the contrast for *Scelio* and *Baryconus*, and might not be enough to trigger a clear response from the spider. In future research it would be interesting to use lures simulating additional wasp features, as well as dead scelionid wasps as lures.

In our experiments with predators, we used three types of *Lyssomanes* spiders: field-collected adults, field-collected juveniles, and lab-reared (captive) adults. In order to consider warning signals more generally, we integrated background contrast, predator vision, predator life stage, predator and prey sizes, and prior opportunities for learning by the predator [3, 5]. Predator origin (being a field adult) was the only significant factor that increased the main probability of detecting vs attacking and avoiding vs attacking. If learning plays a role in avoiding aposematic prey, one would expect field-collected adults to show the greatest discrimination between black and BOB wasps, since the use of lab-reared specimens eliminates the role of learned generalization [31], and field-collected juveniles have had less time to learn. The results of our experiments therefore provide some evidence for learning, an unsurprising finding since learning is common in arthropods [32]. Moreover, the only instances of spiders consuming wasps (non-aposematic, black wasps) were of *Lyssomanes* that had been captured in the field.

A possible innate effect is evident in the clear aversion by captive adults to wasps with the BOB coloration. Evidence for innate aversion has been observed in naive chickens presented with hibiscus harlequin bugs [33], in naive mantises presented with ants [34] and even with ant-mimicking spiders [35], and in jumping spiders presented with army ants [36] or with prey of different colors [37]. Furthermore, it has been shown that many predators are very conservative in their prey preferences, and avoid brightly colored prey even when they are edible [33]. Nonetheless, finding evidence for innate aversion of captive adult spiders to wasps with the BOB coloration does not rule out the possibility that learning is important in this predator prey interaction; indeed, aversions to particular prey colors can change with experience [38]. We found that spiders from natural conditions showed increased aversion over time, and this aversion coincides with the toxicity analyses and the aposematic pattern of the wasps.

## Spectral components of the BOB pattern

Although orange is usually present in aposematic coloration and warning signaling [3, 34], our results show that not all BOB specimens were treated similarly by the spider predators. The orange coloration in BOB wasps varies between genera, despite appearing similar to the human eye. It is possible that the difference between the two light absorbing elements (black

and orange pigments) within each genus of wasps is important in affecting predator responses. Such differences could favor the increase of conspicuousness and consequently intervene in initial avoidance [39, 40].

In general, the dispersion of the reflectance data taken from reference [2] prevent us from making strong assertions, but some features or tendencies can be pointed out from the comparison between the contrast absorption parameter and the results of experiments with live prey summarized in Fig 6. As discussed in reference [2], the dispersion in those measurements derives from several factors, including the fact that for this study, species were grouped into genera. As mentioned previously, the reflectance curves of *Baryconus* have the least dispersion. Therefore, the estimated error in the contrast parameter is the lowest between the genera studied. Nevertheless, the contrast parameter resulted in a value that is within the error, meaning that black and orange cuticle segments have the same absorption. This is consistent with the fact that *Baryconus* presented the higher percentage of "attack" events, suggesting that if color perception is the only variable considered, in this case black and orange are not conspicuous from the point of view of the spider. *Scelio* and *Macroteleia* have higher absorption contrast, and higher orange contribution, suggesting that in those cases the spider should be more likely to differentiate between black and orange. In fact, no experiment involving *Macroteleia* wasps resulted in an "attack" event, all of them fell into the category "detect". The results for *Chromoteleia* are similar in the sense that all the events were distributed between the categories "detect" and "avoid", and the contrast parameter is the second highest in terms of absolute value. Nevertheless, the standard deviation of reflectance measurements is the highest among the genera considered, resulting in an absorption parameter that is within the error estimation. Even though the relationship between the BOB pattern and the behavior of the predator should be taken with caution because of the estimated error in the calculations, we hypothesize that the contrast difference between black and orange from the point of view of the predator visual capabilities is an important factor to be taken into account in behavioral experiments.

It has been reported that jumping spiders are less sensitive to the orange end of the spectrum and thus perceive orange cuticles as more achromatic than structures with relatively more reflectance in the green portion of the spectrum [6]. The aforementioned, plus the absence of a red receptor in many spiders, could cause orange objects to be perceived as monochromatically "green" objects, difficult to distinguish chromatically from green foliage [40–42]. Our results seem to be consistent with observations that arthropods may be able to perceive long wavelength difference via achromatic or luminance information [43], and that some color patterns may function as multicomponent signals [44, 45].

## Conclusion

Most studies of aposematism have used vertebrate predators and larger prey, whereas arthropod predators of smaller prey have generally been neglected [37]. The BOB pattern is extremely widespread in small parasitoid wasps (as well as in a few other insects) and has evidently evolved independently on numerous occasions [1]. To the best of our knowledge, the present study provides the first evidence that this common color pattern might have an aposematic function in these small insects, although further research is needed to confirm this.

There are still many unanswered questions regarding the black-orange-black color pattern found in scelionids and other small wasps. For example, the chemical identity of the pigments and of the putative noxious compounds are still unknown. How does learning interact with variability in the orange tones and the putative toxins? Finally, it would be instructive to examine the spectral properties of other wasps with this color pattern, as well as the visual capacity and responses of other potential predators.

## Supporting information

**S1 File.**
(DOCX)

## Acknowledgments

We give special thanks to Geovanna Rojas Malavasi, Natalia Jiménez Conejo, Paulina Morales Vargas and Anthony Ulate for their support in field data collection, maintenance and rearing of spiders. We also thank Mauricio Valverde Arce for his support in macrophotography and G.B Edwards for his valuable help in the identification of the salticid spiders. We thank our filming crew, Valeria Romero and Soren Pessoa of MANDUCA productions, for their crucial collaboration. We thank two anonymous referees whose comments greatly improved the quality of the manuscript.

## Author Contributions

**Conceptualization:** Rebeca Mora-Castro.

**Formal analysis:** Rebeca Mora-Castro, Marcela Alfaro-Córdoba, Marcela Hernández-Jiménez.

**Funding acquisition:** Rebeca Mora-Castro.

**Investigation:** Rebeca Mora-Castro.

**Methodology:** Rebeca Mora-Castro, Marcela Alfaro-Córdoba, Marcela Hernández-Jiménez, Mauricio Fernández Otárola, Michael Méndez-Rivera, Didier Ramírez-Morales, Carlos E. Rodríguez-Rodríguez, Andrés Durán-Rodríguez, Paul E. Hanson.

**Project administration:** Rebeca Mora-Castro.

**Resources:** Rebeca Mora-Castro.

**Software:** Marcela Alfaro-Córdoba, Marcela Hernández-Jiménez.

**Supervision:** Rebeca Mora-Castro.

**Validation:** Rebeca Mora-Castro.

**Visualization:** Rebeca Mora-Castro, Marcela Alfaro-Córdoba, Marcela Hernández-Jiménez, Mauricio Fernández Otárola, Michael Méndez-Rivera, Didier Ramírez-Morales, Carlos E. Rodríguez-Rodríguez, Andrés Durán-Rodríguez, Paul E. Hanson.

**Writing – original draft:** Rebeca Mora-Castro, Marcela Alfaro-Córdoba, Marcela Hernández-Jiménez, Mauricio Fernández Otárola, Michael Méndez-Rivera, Didier Ramírez-Morales, Carlos E. Rodríguez-Rodríguez, Andrés Durán-Rodríguez, Paul E. Hanson.

**Writing – review & editing:** Marcela Alfaro-Córdoba, Marcela Hernández-Jiménez, Mauricio Fernández Otárola, Michael Méndez-Rivera, Didier Ramírez-Morales, Carlos E. Rodríguez-Rodríguez, Paul E. Hanson.

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
