## [Decision Letter · Decision Letter 0]

25 Aug 2020

PONE-D-20-22240

First evidence for an aposematic function of a very common color pattern in small insects.

PLOS ONE

Dear Dr. Mora-Castro,

Thank you for submitting your manuscript to PLOS ONE. After careful consideration, we feel that it has merit but does not fully meet PLOS ONE’s publication criteria as it currently stands. Therefore, we invite you to submit a revised version of the manuscript that addresses the points raised during the review process.

As you will note in the detailed comments below, there was consensus among all three reviewers that the results are compelling and should be published. In particular, there was enthusiasm for work on aposematism being tested in a non-vertebrate system (as the authors note, this is a common pattern in insects) and the considerable effort undertaken to produce these data. However there is also universal agreement that significant changes should be made prior to publication. The most vital suggested revisions relate to improving the clarity of the methods and reporting appropriate statistical values. There are also comments from each reviewer that suggest improvements in presenting the rationale for your experimental design, describing crucial results, and acknowledging potential limitations regarding what is known about scelionid biology. Please consider all specific comments made by the reviewers carefully and incorporate this feedback into your next submission. If the authors disagree with any particular suggestions or comments, these disagreements should be noted and justified in a response letter.

We look forward to receiving your revised manuscript.

Kind regards,

Phillip Barden

Academic Editor

PLOS ONE

Journal Requirements:

Reviewers' comments:

Reviewer's Responses to Questions

**Comments to the Author**

1. Is the manuscript technically sound, and do the data support the conclusions?

Reviewer #1: Yes

Reviewer #2: Partly

Reviewer #3: Partly

2. Has the statistical analysis been performed appropriately and rigorously? 

Reviewer #1: I Don't Know

Reviewer #2: No

Reviewer #3: Yes

3. Have the authors made all data underlying the findings in their manuscript fully available?

Reviewer #1: Yes

Reviewer #2: No

Reviewer #3: Yes

4. Is the manuscript presented in an intelligible fashion and written in standard English?

Reviewer #1: Yes

Reviewer #2: Yes

Reviewer #3: Yes

5. Review Comments to the Author

Reviewer #1: Review

This is a novel approach to a complex question, and there is little doubt of the efforts made to obtain these data, even if they are ‘noisy’ (biology is noisy!). I think that the material is interesting and should be published, but as written the rationale behind certain tests is a bit unclear and the writing could be streamlined so as not to lose the reader. Please note that I have replied "I don't know" about the statistics primarily because I lack the skills to evaluate the colour-based information.

The introduction is very well-written and the referencing is very good. The methods are equally thorough! I can imagine, if there is a need to shorten the manuscript, that considerable (and commendable) detail from this section could be removed.

L. 46. Define BOB on first mention

Methods

Detailed comments below. The methods are extremely detailed, and yet I am confused, because there is reference that animations were used, but there is no detail on this at all, so in the end, I am forced to think it was using live prey and rice. What is the situation with reference to animations (specifically “Editing was done in Adobe Premiere, the animations and layout in Adobe After Effects”)? Also, I suggest the authors look at references by Cynthia Tedore:

https://academic.oup.com/beheco/article/26/2/510/258466

https://jeb.biologists.org/content/216/9/1744.short

L. 174. Change “in” for “at” 60 fps

L. 174-175. As you have not yet discussed animations, this information comes out of the blue- so far all we have read is about live-prey capture observations. Move to next section? .

L. 199. This section is very detailed, yet we are unclear of the overview of the test or why it was performed. Before going into details of the spectral details of the paint used on rice, we need a clear understanding of how the rice, the spider, and the arena ‘interacted’. How was the rice made to move? Did it float? Based on the comment in L 175, I half expected it to be an animation of rice. At this point I am getting thoroughly lost. The view of the forest is being lost to the detail of the trees, so to speak. The first thing to do is move any description of the automated cage up, so that the reader can understand the arena before these details are discussed.

L. 256, add comma after “prey”

L. 310. How was the dragline observed? And why?

L. 323. How did you define “similarly” versus “differently”?

Results

It is striking that actually there are very few differences between spider category and wasp pattern colour overall…

In experiments of spiders with live prey you refer to “captive-adults”, which I think belong in the category previously referred to as “reared”. Please use expressions consistently.

I am sure Fig 3 could be a supplementary figure, if needed, as could Fig 5 (possibly 8, too?).

L. 401. Are you sure they only have a dragline when attacking? Many salticids deposit a dragline while walking.

I am not an expert in colour analysis, but I thought the approach taken in Fig 7 was imaginative and helpful. However, I was a bit lost about the rationale for the bioluminescence testing.

Discussion

L. 591. You are correct is saying that learning of aposematic signals is often assumed. Nevertheless, this need not be the case (theoretically or in practice), as I have argued at length, demonstrating innate aversion of ants by jumping spiders (and mantises) on numerous occasions:

Nelson, X.J., Jackson, R.R., Li, D., Barrion, A.T., Edwards, G.B., 2006. Innate aversion to ants (Hymenoptera : Formicidae) and ant mimics: experimental findings from mantises (Mantodea). Biological Journal of the Linnean Society 88, 23-32.

Nelson, X.J., Jackson, R.R., 2006. Vision-based innate aversion to ants and ant mimics. Behav Ecol 17, 676-681.

https://link.springer.com/article/10.1007/s10164-020-00639-1

Reviewer #2: The manuscript entitled “First evidence for an aposematic function of a very common color pattern in small insects” tested whether a pattern coloration (black-orange-black) common in parasitoid wasps functions as an aposematic signal to indicate toxicity to small arthropod predators. I believe it is a novel test of the idea of aposematism (aposematic coloration has been always tested for vertebrate predators, as the authors state), and I found the results very interesting and worth publishing. However, I believe the manuscript could be polished further. In general, I believe the presentation of results could be more clear, and free from phrases that belong to discussion and methods; that some details of the methods could be included as Supplementary Materials; and, that the Discussion could be strengthened by adding more literature.

Specifically, these are my major recommendations:

1. The abstract should include more results and less details of the methods. The last sentence of the abstract (L-32-34) is uninformative.

2. L. 81-86. If you list the predictions in the same order than the results, that facilitates reading. Also, the prediction of a greater contrast (between black and orange) triggering fewer attacks is not explained here.

3. Right below the results there is a section called “experiments with live prey”, that is not line-numbered, and that belongs to the Methods; indeed, some parts are repeating information already present in the methods. Fig 3 should be presented for the Methods section.

L. 379-385. Again, most of this paragraph should be in Methods.

L. 395-407. Here, you reader is dying to know whether black wasps were more attacked than BOB wasps, but that is not straightforward to find in this paragraph. In fact, in one part tell us that the wasp color was not significant, but at the end it says that field adults only attacked black wasps… it is confusing. Also, were all wasp genera combined here?

It is not clear if the statistical model (Fig. 4) is accounting for repeated measures, i.e. including the spider ID as a block or random factor in the model.

L. 408-428. This whole page is missing associated statistical values.

L. 442-443. This result about Baryconus spectral components of orange and black makes me want to see the probability of attack vs avoid vs detect separated by genus.

L. 448-458. This section is again missing statistical tests. If the absorption contrast is a continuous variable, can you run a model of whether the absorption contrast influences the probability of attacking vs avoiding only for those spiders that detected the wasps?

Fig 7. The percentage is calculate over which total?

Table 1. How is same and different calculated here?

L. 502-514 is missing stats.

Table 2. Please consider presenting this information in a figure.

L. 560-567. Also consider that the paint was matching the genus with the least differences between orange and black. Is there any literature (even in vertebrates) where painted inanimated objects served to test aposematic signals? Perhaps you want to include that. What happened with the black vs white walls on the arena?

Discussion: please consider including more literature, for example about field vs captivity-reared animals regarding familiarity with aposematic signals,

Minor comments:

L 55-56. Is there any report of jumping spiders preying on parasitoid wasps? If so, please include the reference to support the selection of this spider as predator.

L. 92. Please provide m.a.s.l for that site.

L. 103-107. You can say here that all four genera have BOB and black specimens, so the reader doesn’t have to guess.

L. 115-123. Are the four genera of wasps also found at this locality?

L. 133-146. This section could be shortened if you remove some irrelevant details. Also, some details of the “Salticid rearing” section could be moved to Supplemental material.

L. 174 – which animation?

L. 176-197: These bullet points can be redacted in a paragraph, and some of the points could be shortened as they are too wordy. For example, point 6 could simply be: Each spider was allowed 60 s of acclimatization in the chamber, before introducing the prey.

L. 212-213. Was the reflectance of the paint measured on painted grains of rice, or on which surface?

L. 199-249: Many of the details on the selection of the paint should be moved to Supplementary Material.

L 315-316. Please explain this section better (frequency according to time). Why do you have 5 min and 10 min time slots mixed?

L. 443-447. These sentences are methods.

L. 476-483. This should be considered for methods and as Supplementary Material.

Fig. 8. Can you please explain why the paints were compared to the Baryconus only. Considering that you also reported that the black and orange of this genus had similar spectral components, could that influence your results with the lures?

L. 503. LC50 should be a concentration that kills 50% of the animals, but it is a percentage, why?

Table 2: IC? Or CI?

Reviewer #3: The topic of the manuscript is interesting, as it focuses on a phenomenon described only in fairly large animals, but common among the small arthropods (not only insects). Aposematic function of colour patterns was previously tested in much larger animals (e.g. medium and large insects, reptiles and amphibians). There are three major reasons for that: 1) it is much easier to use larger animals for tests; 2) aposematism was earlier reported from such large animals and 3) if aposematism has evolved in an organism there should be a receiver (predator) the feature is advertised to. Typically, such predatory animals were assumed to be animals larger than the prey, with high visual capacity – high enough to perceive the signal (vertebrates, typically birds). This is indeed the first study that exploits minute predator with high visual capacity in the context of interactions with BOB-patterned aposematic prey.

The model spider seems properly selected (common, easily identifiable predator). The insects used in the experiments seem, however, less convincing as prey models and therefore they should be described in more detail. Our knowledge about tropical scelionids is still very scanty. Due to the lack of taxonomic knowledge the authors identified the species of wasps from 1 out of 4 genera. This can potentially create some problems in interpretation, as similar species of scelionids may have different behaviour and chemical constitution and although salticids are mostly visual hunters they also exploit other senses in prey capture. It is, therefore, possible that other than visual cues might have been exploited during interactions with different live as well as coloured false prey. I personally admire the workload and time sacrificed to collect the insects in the field. Still the fact that we don’t know exactly the actual material as much as we would like to should be critically discussed.

Description of some methods seems too detailed (e.g. concerning salticid and their egg sac collection), while other methods, which seem important to understand and interpret the results, are scanty (e.g. the behaviour of spiders and scelionids in the experimental chambers, motion pattern of false prey).

In the results there seem to be statements without statistical support (and a name of a test) e.g. lines 420-428, 451-453 referring to Fig. 7. Again in lines 490-492 referring to Table 1 there is information that the data were “…tested for independence, and there was no evidence of dependence in any 492 of the cases”, but no results and the tests used were given. Similarly in the discussion (lines 532-535) conclusions are drawn without a reference to a statistical test. Appropriate tests should be carried out or shown to check the statements.

In the study “the use of silk” by the spiders was included in the analyses (e.g. Table 1, lines 209, 310). It is not explained, however, why just this variable was included. In lines 401-402 it is also mentioned that “these spiders use silk only when attacking”. This seems unlikely for a vegetation-dwelling spider, which typically should exploit draglines in locomotion and jumps (including jumps onto prey).

6. PLOS authors have the option to publish the peer review history of their article (what does this mean?). If published, this will include your full peer review and any attached files.

Reviewer #1: No

Reviewer #2: No

Reviewer #3: No

---

## [Author Response · Author response to Decision Letter 0]

24 Sep 2020

In the attached files, we include a 19-page PDF (Response to reviewers) where we respond in detail to all the comments of the three reviewers. We appreciate the contributions and clearly the manuscript improved considerably.

---

## [Decision Letter · Decision Letter 1]

6 Nov 2020

PONE-D-20-22240R1

First evidence for an aposematic function of a very common color pattern in small insects.

PLOS ONE

Dear Dr. Mora-Castro,

Thank you for submitting your manuscript to PLOS ONE. After careful consideration, we feel that it has merit but does not fully meet PLOS ONE’s publication criteria as it currently stands. Therefore, we invite you to submit a revised version of the manuscript that addresses the points raised during the review process.

I thank the authors for their revision and patience with the second review round. Because it was not possible to secure sufficient reviews for a second revision, I acted as a secondary reviewer here. Based on Reviewer 1 comments and my own assessment, another round of revision is necessary. Although the paper reads much more clearly throughout the introduction and methods, the results and key figures still need substantial revision. Please find my comments and those of Reviewer 1 below.

Major comments:

Figs 3 and 4 are very difficult to interpret and should be heavily revised. I, along with Reviewer 1, am having a very difficult time understanding where these data come from and what they are suggesting relative to the authors’ conclusions – particularly Figure 3. The two figures also appear to be contradictory. An additional concern is that there are no units on the axes and that Figure 3 caption is very brief. Add units to the axes in both figures if they are to be kept in their current form and otherwise revise to make the results clearer as suggested by Reviewer 1.

Fig 4, why is this a stacked bar graph? This makes it much more difficult to directly compare BOB and Black wasp categories. In addition, I recommend including a label for the “detect” “attack” and “avoid” panels, as well as noting how the observations were binned in this scheme in the figure caption.

Line 495: "no spider consumed a wasp with BOB coloration." this appears to be among the more compelling evidence for aposemetism, yet results are not presented except when combined another "attack" behavior. Could these data be presented to support this sentence in the discussion?

The toxicity results appear to be the most indicative of aposematism, however, the behavioral data, at least as presented here, are not conclusive (among field adults, black wasps are avoided more frequently, BOB wasps attacked more frequently in most timeframes according to Fig 4). Given this, I do not feel the final sentence of the abstract “By combining the results from the three types of experiments, together with a statistical analysis, we confirmed that the BOB color pattern plays an aposematic role.” is warranted. Please revise this conclusion to be more equivocal (e.g. “some results suggest”), given that the data do not definitively support this conclusion.

Minor comments:

line 245: should be "among" instead of "between" as there are more than two genera

line 257: "criterium" should be "criterion"

We look forward to receiving your revised manuscript.

Kind regards,

Phillip Barden

Academic Editor

PLOS ONE

Reviewers' comments:

Reviewer's Responses to Questions

**Comments to the Author**

1. If the authors have adequately addressed your comments raised in a previous round of review and you feel that this manuscript is now acceptable for publication, you may indicate that here to bypass the “Comments to the Author” section, enter your conflict of interest statement in the “Confidential to Editor” section, and submit your "Accept" recommendation.

Reviewer #1: (No Response)

2. Is the manuscript technically sound, and do the data support the conclusions?

Reviewer #1: Partly

3. Has the statistical analysis been performed appropriately and rigorously? 

Reviewer #1: I Don't Know

4. Have the authors made all data underlying the findings in their manuscript fully available?

Reviewer #1: Yes

5. Is the manuscript presented in an intelligible fashion and written in standard English?

Reviewer #1: Yes

6. Review Comments to the Author

Reviewer #1: Dear Authors,

many of the issues previously raised have been addressed in this current revision. However, unfortunately, I still find myself a bit confused. The methods and much more succinct and are much easier to follow in the current form, but more work is needed in some areas. I have outlined both minor and mojor comments below.

Major:

L. 298, I still think “in the same way” and “in a different way” is pretty vague and no better than the previous “similarly” and “differently”; some description of precisely what is meant here is really necessary, in my view.

L. 544-546. The relationship between figs 3 and 4 needs clarification (in the results), as some of the results and discussion appear somewhat contradictory. For example, in the discussion, you state “Moreover, field-collected adults were more likely to detect 544 or avoid than to attack, compared to captive adults which appeared to exhibit a more limited behavioral scale with attack being the most frequent”. However, a quick look at Fig 4, which uses the same axes for each row of panels, suggests differently. In part this is because it is expressed in real numbers, not as percentages. However, fig. 3 suggests that field adults NEVER attacked orange wasps, yet fig 4 strongly suggests otherwise. I think these data are from the same experiment yet partitioned differently. Consequently, I am confused about what the results actually were. If I am mistaken in my interpretation, then the explanation for these discrepancies needs to be made crystal in the results and discussion. Alternatively, there is some problem with the classification of the data?

L. 60-63. Technically, whether a spider is di, tri or tetrachromatic is not the cause of high spatial acuity; this is due to photoreceptors in the retina being packed densely in arrays that function as light guides. Perhaps re-work carefully, as colour vision does not equal high acuity, and the causes behind each are different.

Minor:

L. 83-89. Please number your predictions

L. 181, add comma after “used”

L. 186, add comma after “arena”

L. 187, add comma after parenthesis

L. 273, add comma after “out”

L. 370. Can you provide a reference for “spiders produce more silk when attacking”; this is news to me.

L. 418, remove “a” before “previous”

7. PLOS authors have the option to publish the peer review history of their article (what does this mean?). If published, this will include your full peer review and any attached files.

Reviewer #1: No

---

## [Author Response · Author response to Decision Letter 1]

7 Dec 2020

Among the various documents that we attach to this online system for review, we include a document entitled -response to reviewers- in which we answer all the reviewers' comments and incorporate all the suggestions submitted by them.

---

## [Decision Letter · Decision Letter 2]

23 Dec 2020

PONE-D-20-22240R2

First evidence for an aposematic function of a very common color pattern in small insects.

PLOS ONE

Dear Dr. Mora-Castro,

Thank you for submitting your manuscript to PLOS ONE. After careful consideration, we feel that it has merit but does not fully meet PLOS ONE’s publication criteria as it currently stands. Therefore, we invite you to submit a revised version of the manuscript that addresses the points raised during the review process.

Your revised manuscript has been reviewed by an external reviewer and myself. The consensus is that the manuscript is substantially improved and nearing suitability for publication, therefore I am recommending an additional, potentially final, round of minor revision. Below you will find comments from the external reviewer. Please consider these carefully and incorporate them into your next submission. In particular, I encourage you to include Table S4 in the main text and revise your bar graphs to remove the "full width" bars when one category is absent. 

We look forward to receiving your revised manuscript.

Kind regards,

Phillip Barden

Academic Editor

PLOS ONE

Reviewers' comments:

Reviewer's Responses to Questions

**Comments to the Author**

1. If the authors have adequately addressed your comments raised in a previous round of review and you feel that this manuscript is now acceptable for publication, you may indicate that here to bypass the “Comments to the Author” section, enter your conflict of interest statement in the “Confidential to Editor” section, and submit your "Accept" recommendation.

Reviewer #1: All comments have been addressed

2. Is the manuscript technically sound, and do the data support the conclusions?

Reviewer #1: Yes

3. Has the statistical analysis been performed appropriately and rigorously? 

Reviewer #1: Yes

4. Have the authors made all data underlying the findings in their manuscript fully available?

Reviewer #1: Yes

5. Is the manuscript presented in an intelligible fashion and written in standard English?

Reviewer #1: Yes

6. Review Comments to the Author

Reviewer #1: This is a much improved manuscript and I only have minor revisions suggested. However, I still think the final conclusion (L. 620) should be a bit more circumspect, given your results, and I also think the discussion could use a few more sentences to give it more depth. Specifically, you talk about innate aversion in chicken and mantises, yet do not mention work on jumping spiders:

Nelson, X.J., Aguilar-Arguello, S., Jackson, R.R., 2020. Widespread army ant aversion among East African jumping spiders (Salticidae). Journal of Ethology 38, 185-194.

Nelson, X.J., Jackson, R.R., 2006. Vision-based innate aversion to ants and ant mimics. Behav Ecol 17, 676-681.

Additionally, there is relevant research on colour aversion in jumping spiders which should also be added to your discussion:

https://onlinelibrary.wiley.com/doi/full/10.1111/eth.12859

Taylor, L. A., Maier, E. B., Byrne, K. J., Amin, Z., & Morehouse, N. I. (2014). Colour use by tiny predators: Jumping spiders show colour biases during foraging. Animal Behaviour, 90, 149–157. https://doi.org/10.1016/j.anbehav.2014.01.025

Other comments:

Graphs 3 and 4. The double width bars when there are no ‘black’ data are a bit confusing. Make the orange bar the same width as the others, and simply not have a ‘black bar’ where there are no data.

I think Table S4 is definitely worth putting in the body of the text. To me, this partitioning makes more sense than the ones used in the graph, and shows things more clearly.

7. PLOS authors have the option to publish the peer review history of their article (what does this mean?). If published, this will include your full peer review and any attached files.

Reviewer #1: No

---

## [Author Response · Author response to Decision Letter 2]

19 Jan 2021

Among the documents attached to the system is the document entitled: Response to reviewers where we respond one by one to all the reviewers' suggestions.

---

## [Editor Report · Decision Letter 3]

28 Jan 2021

First evidence for an aposematic function of a very common color pattern in small insects.

PONE-D-20-22240R3

Dear Dr. Mora-Castro,

We’re pleased to inform you that your manuscript has been judged scientifically suitable for publication and will be formally accepted for publication once it meets all outstanding technical requirements.

Kind regards,

Phillip Barden

Academic Editor

PLOS ONE
---

## [Editor Report · Acceptance letter]

1 Feb 2021

PONE-D-20-22240R3 

First evidence for an aposematic function of a very common color pattern in small insects. 

Dear Dr. Mora-Castro:

I'm pleased to inform you that your manuscript has been deemed suitable for publication in PLOS ONE. Congratulations! Your manuscript is now with our production department. 

Kind regards, 

on behalf of

Dr. Phillip Barden 

Academic Editor

PLOS ONE